# Competitive Growth of Sulfate-Reducing Bacteria with Bioleaching Acidophiles for Bioremediation of Heap Bioleaching Residue

**DOI:** 10.3390/ijerph17082715

**Published:** 2020-04-15

**Authors:** Aung Kyaw Phyo, Yan Jia, Qiaoyi Tan, Heyun Sun, Yunfeng Liu, Bingxu Dong, Renman Ruan

**Affiliations:** 1CAS Key Laboratory of Green Process and Engineering, Institute of Process Engineering, Chinese Academy of Sciences, Beijing 100190, China; engr.aungkyawphyo@gmail.com (A.K.P.); qytan@ipe.ac.cn (Q.T.); sunhy@ipe.ac.cn (H.S.); bxdong@ipe.ac.cn (B.D.); rmruan@ipe.ac.cn (R.R.); 2National Engineering Laboratory for Hydrometallurgical Cleaner Production Technology, Chinese Academy of Sciences, Beijing 100190, China; 3College of Chemistry Engineering, University of Chinese Academy of Sciences, Beijing 100049, China; 4Department of Mines, Ministry of Natural Resources & Envrionmental Conservation, Nay Pyi Taw 15011, Myanmar; 5State Key Laboratory of Biochemical Engineering Institute of Process Engineering, Chinese Academy of Sciences, Beijing 100190, China; 6Wanbao Mining Limited, Beijing 100053, China; liuyf@wbmining.com

**Keywords:** acid mine drainage, bioremediation, heap bioleaching, Iron- and sulfur-oxidizing microbes, source control, sulfate-reducing bacteria

## Abstract

Mining waste rocks containing sulfide minerals naturally provide the habitat for iron- and sulfur-oxidizing microbes, and they accelerate the generation of acid mine drainage (AMD) by promoting the oxidation of sulfide minerals. Sulfate-reducing bacteria (SRB) are sometimes employed to treat the AMD solution by microbial-induced metal sulfide precipitation. It was attempted for the first time to grow SRB directly in the pyritic heap bioleaching residue to compete with the local iron- and sulfur-oxidizing microbes. The acidic SRB and iron-reducing microbes were cultured at pH 2.0 and 3.0. After it was applied to the acidic heap bioleaching residue, it showed that the elevated pH and the organic matter was important for them to compete with the local bioleaching acidophiles. The incubation with the addition of organic matter promoted the growth of SRB and iron-reducing microbes to inhibit the iron- and sulfur-oxidizing microbes, especially organic matter together with some lime. Under the growth of the SRB and iron-reducing microbes, pH increased from acidic to nearly neutral, the Eh also decreased, and the metal, precipitated together with the microbial-generated sulfide, resulted in very low Cu in the residue pore solution. These results prove the inhibition of acid mine drainage directly in situ of the pyritic waste rocks by the promotion of the growth of SRB and iron-reducing microbes to compete with local iron and sulfur-oxidizing microbes, which can be used for the source control of AMD from the sulfidic waste rocks and the final remediation.

## 1. Introduction

Mining activities produce large volumes of waste, comprising mainly sub-economic waste rock and tailings. Among these wastes, most of them contain sulfide minerals, especially pyrite. When exposed to the atmosphere and especially after the natural growth of autotroph iron- and sulfur-oxidizing microorganisms, the oxidation of sulfide mineral accelerates, generating a huge amount of acid mine drainage (AMD) through oxidation [1,2,3]. Drainage from mining activities is often acidic and also frequently associated with high concentrations of heavy metals and metalloids. It is considered to be one of the most important sources for heavy metal contamination in the environment [4].

When exposed to the atmosphere, sulfide minerals in the waste rocks are oxidized under water and oxygen, and then generate acid, make the environment of the waste rock acidified. The dissolution of sulfide minerals relies to a large extent on the chemical and microbiological processes caused by the reaction of the oxidant Fe^3+^ and the activity of natural iron- and sulfur-oxidizing microorganisms [5]. The Fe^3+^ is more aggressive and effective than O_2_ for pyrite oxidation in acidic conditions [6,7,8,9]. Iron sulfide minerals such as pyrite usually provide the source of oxidant Fe^3+^. Pyrite usually acts as the most important source of AMD, since it is the most abundant sulfide minerals on earth [10]. 

SRB bacteria are usually employed to treat an AMD solution by growing SRB in the AMD solution, through the microbial process of metal- and sulfate-reducing, and producing alkalinity, and attenuating the movement of metals by the precipitation of sulfide minerals [11,12]. These processes are exploited in ex situ treatment of AMD after it comes out from the rocks [13,14,15]. However, a few researchers have used the SRB directly to treat the acid-generating waste rocks in situ to inhibit the oxidation of sulfide minerals and to trigger the co-precipitation of metal with sulfide, for source control of the AMD, and further rehabilitation of the waste rocks. It was found that Fe(III)- and sulfate-reducing microbes co-existed with iron- and sulfur-oxidizing microbes in the tailing dams [16,17], suggesting that biogeochemical Fe and S cycling existed in the natural acid condition [3]. Although tailings were usually fine and low in oxygen concentration inside the dam, it was still always the predominance of oxidation processes that resulted in the acidifying of the waste rock [16,17] rather than the reducing processes. Providing a suitable condition for SRB is really needed to promote the growth of reducing microbes over the oxidizing microbes. 

Mining activities generate waste, including waste rock, tailings, and heap leaching residue. All around the world, acid-generating wastes are produced by about billions of tons every year, and the generation of acid mine drainage has been the most important source of the heavy metal pollution around the world. Among the acid-generating wastes, the heap leaching residue is generated every year by hundreds of millions of tons. It is the most difficult to remedy and to revegetate because, during the heap bioleaching process, the whole heap is completely acidified, usually under pH < 1.5, and the growth of iron- and sulfur-oxidation microbes is also promoted during the bioleaching process. If the SRB remediation of the heap leaching residue succeeds, it will be easier for remediation of other acid-generating waste rocks. In this study, the heap bioleaching residue is used to study the bioremediation process of SRB and iron reducing microbes for competition with the local autotrophic bioleaching microbes under the acidic condition, the SRB community succession during the remediation, and the influencing factors, for implication to the source control of AMD from the acid-generating rocks in situ.

## 2. Materials and Methods

### 2.1. Sample Collection and Characterization 

Heap bio-leached residues were sampled from 3 heap leaching cells at Monywa copper mine, Myanmar, in 2018. The heap bioleaching plant is in a tropical climate area with an annual temperature of 20–40 °C. The copper ore before heap bioleaching consists of about 9% pyrite, about 0.8% copper sulfides (mainly chalcocite), 50% silica, and other gangue minerals, with very few alkaline gangues. Heap leaching is stacked with run-of-mine, and pyrite oxidation results in excessive acid and iron accumulation in the cycling solution during the leaching. The microbial community in heaps is mainly the bioleaching acidophiles in the genus of *Ferroplasma, Acidithiobacillus, Leptospirrilium,* as previously reported in Jia et al. [18].

After heap bioleaching, the residue is still uncovered and exposed weathering, and generates acid mine drainage. Leach solution from the bottom of the cell was collected and the cation ions were assayed by atomic absorption spectrometry (AAS) (ZEEnit@700P, Analytik Jena, Germany) and inductively coupled plasma mass spectrometry (ICP–MS) (7700, Agilent Technologies, Santa Clara, CA, USA). Anion ions were assayed by ion chromatography (IC) (ICS600, PerkinElmer, Waltham, MA, USA), free acid was assayed by titration, and ionic strength was assayed by precipitating all elements and getting the total weight (Table 1). The leaching cells were usually 30–40 m high, with a total residue of 150 million tonnes accumulated over about 20 years. Twenty-seven samples from three cells were taken for the test work. The element content was assayed by ICP–MS, and net acid generating potential was assayed by H_2_O_2_ oxidation and NaOH consumption. The content of total sulfur is 6.42% on average, and the sulfide sulfur content is 3.81% (mainly pyrite, few unleached chalcocites) with the potential to generate 77.8 kg t^−1^ of sulfuric acid on average (Table 2). Cu in the residue is about 0.07% on average and with not too many other heavy metals.

### 2.2. Culture of Iron- and Sulfate-Reducing Bacteria

The incubation solutions for iron- and sulfate-reducing bacteria were the AMD solution from Monywa heap leaching residue. Local mine impacted anaerobic soil (pH about 4.0) was also collected near the acid mine drainage. Anaerobic soil was stirred, and the filtrate solution was collected. AMD solutions are adjusted by NaOH to different pH of 2.0 and 3.0. Then, each 100 ml AMD solution under different pH was added with the anaerobic soil solution 5 ml, 2 g/L yeast powder, 2 g/L sodium lactate, and 2 g pure pyrite. The anaerobic bottles were kept in the anaerobic culture box under the control temperature of 35 °C. During the culture, Fe^2+^ concentration in the bottles was measured weekly. The microbes in the solution were collected after 60 days of incubation, and then the microbial community was analyzed by amplifying 16S rDNA and tested using the MiSeq high-throughput sequencing technique. Additionally, the two microbial consortiums after cultivation at pH 2.0 and 3.0 were used as the inoculation SRB in the column tests.

### 2.3. Column Tests

The heap leach residue was crushed by a jaw crusher to a size of about P_80_ = 20 mm, for column incubation tests. Subsequently, its size distribution were assayed by standard sieves (10, 5, 2, 1, and 1/2 inch; 100 and 200 mesh) and Cu content for each size was assayed. Five samples were prepared with the same size distribution, by weighing and mixing of each size fraction. One of which was pulverized for chemical assay and four samples were used for column tests.

Four acid-proof 316 stainless-steel columns (100 cm high and 10 cm inner diameter with a rubber liner) with a water jacket were used in the column experiment. Each column was charged with the same 6 kg non-sterile (the ore was agglomerated with 4 wt % sterilized water) and then operated at the set temperatures of 35 °C maintained by a water bath. The columns include the 6-kg ore were set: column 1 as the control (control); column 2 with 0.6% lime (lime) added; column 3 with organic matters (yeast powder and sodium lactate, each 1%) (OM) added; column 4 with lime (0.6%) and organic matters (yeast powder and sodium lactate, each 1%) (lime + OM) added. SRB from the pH 2.0 and pH 3.0 anaerobic bottles was added to each column of 10 ml each. A nylon net syringe tube with connecting hole thin tube for the purpose of solution sample collecting was hung in the middle of the column. Water was added to each column and a water level above the ore of 3 cm was kept.

The solution of 10 ml in the columns was collected by syringe and the sample immediately measured for chemical analysis such as pH, Eh, TFe, Fe^2+^, Cu, TOC, and cell numbers weekly; microbial community test was performed at 30, 90, and 150 days. For the microbial community collection, four representative liquid samples (100 mL) were sampled and then immediately collected by 0.22 μm filter paper, for DNA extraction and barcoded pyrosequencing.

At the end of the test, after 180 days incubation, the columns were drained, and solution samples were collected into anaerobic bottles with rubber seals for chemical analysis. Additionally, 3 kg residue sample was dried at 50 °C for 3 days, and then assayed for element contents. The other 3 kg residue was kept frozen −80 °C for DNA extraction and microbial community analysis. 

### 2.4. Chemical Analysis

The solution samples were collected periodically by a syringe-driven filter (100 mesh) buried inside of the column. The pH value was measured by pH meter (Mettler Toledo), and the redox potential of the solution was measured using the Ag/AgCl reference electrode immediately after collection (Mettler Toledo). Microbes number counting in the solution was carried out in a hemocytometer (0.1 mm, 1/400 mm^2^) by a microscope (CX31, Olympus, Japan) with phase contrast condenser and equipped with a Charge Coupled Device (NC7030, Oplenic, China). A 100× plan achromat objective was used to check the cell number. After cell counting, the 100 mL washed solutions per month were filtered through a Millipore filter (0.22 μm). Then the microbes on the filter paper were used for DNA extraction as described in [19]. SO_4_^2−^ was measured by UV spectroscopy devices using the BaSO_4_ to determine sulfate concentration in the samples [20]. 

The concentration of copper and total iron was determined using the ICP-OES (Optima 5300DV, PerkinElmer, Waltham, MA, USA ). Analysis of TOC was conducted using a spectrophotometer (TOC-L, CPH, Shimadu, Japan). Before TOC measurements, samples were acidified with concentrated sulfuric acid and then sparged with nitrogen gas for 5 to 10 min to remove dissolved sulfide.

Residue samples are dried in the oven at 50 °C and ground to a fine powder for assay of copper, iron, total sulfur, and reduce sulfur. The fresh residue was fried at −80°C for DNA extraction and microbial community assay. The proportions of sulfide minerals were determined by a mineral liberation analyzer. 

### 2.5. Microbial Community Analysis 

DNA from the anaerobic bottles and the columns (solutions and residues) are extracted by a FastDNA Spin kit (Bio 101, MP Biomedicals, Santa Ana, CA, USA) according to the manufacturer’s protocol. The F515 (5′-GTGYCAGCMGCCGCGGTAA) and R806 (5′-GGACTACNVGGGTWTCTAAT) primers were used to amplify the bacterial and archaeal 16S RNA genes V4 hypervariable region [21]. PCR reaction was performed in triplicate. Amplicons were excised from 2% agarose gels and purified using the Axyprep DNA Gel Extraction Kit (Axygen Biosciences, Union City, CA, USA) according to the manufacturer’s instructions. Purified amplicons were sequenced using an Illumina MiSeq platform according to the standard protocols. Data were analyzed and quality-filtered using Quantitative Insights Into Microbial Ecology (QIIME) [22]. Operational units (OTU) were clustered at the sequence similarity level of 97% using UPARSE (version 7.1, http://drive5.com/uparse/) and chimeric sequences were identified and discharged using UCHIME. The sequences were compared using BlAST against NCBI. 

## 3. Results and Discussion 

### 3.1. Acidophilic Iron- and Sulfate-Reducing Microbes

The anaerobic incubation of the local AMD and soil microbial community of Monywa copper mine provided the iron- and sulfate-reducing microbial community. During the incubation, Fe^2+^ concentration firstly increased greatly and then decreased (data not shown), showing the reduction of Fe^3+^ to Fe^2+^, then decreased as the Fe^2+^ co-precipitated with the S^2−^, and finally showing the black precipitates and H_2_S smell. Till the end of the incubation at day 60, the pH increased from 2.0 to 3.3, and from 3.0 to 4.7, respectively, in the two bottles, even though pyrite was present in the bottles. The elevated pH in the two bottles suggested the microbial iron- and sulfate-reducing process, and also no oxidization of the pyrite.

The microbial community in pH 2.0 bottle showed the dominant phylum of Archean *Euryarchaeota* and bacteria *Proteobacteria* (Figure 1), of which the Archean *Ferroplasma* species is most abundant (61.8%), and then bacteria *Acidophilium* (23.0%), and also about 5.5% of the sulfate-reducing bacteria *Clostridiaceae*, and very few *Acidithiobacillus* (<1%; Table 3). *Ferroplasma* and *Acidophilium* both grew in the acid condition, with the capacity to anaerobically degrade the organic matter and reduce Fe^3+^ to Fe^2+^. The *Clostridiaceae* here are the species that reduce sulfate to sulfides. It was previously found that the *Ferroplasma* was the dominant microbes in the Monywa bioleaching heaps [18]. In addition, during the incubation in the anaerobic bottle under pH of 2.0, it continued to be the dominant species because of its capacity to undergo autotrophic growth upon ferrous oxidation under aerobic conditions and undergo heterotrophic growth on organic matter under anaerobic conditions [2,23,24]. However, the other autotrophic bioleaching acidophiles, such as *Leptospirillum* and *Acidithiobacillus,* decreased significantly because no oxygen is supplied. The dominant genus in the pH 3.0 bottle was the sulfate-reducing *Clostridiaceae (67.9%)* and *Desulfosporosinus* (18.7%), and *Alicyclobacillaceae* (8.6%) (in *Firmicutes* phylum) (Figure 1, Table 3), showing that higher pH was more suitable for the growth of SRB. 

It is usually suggested that the SRB bacteria favor pH > 5.5. Previous studies also tried to grow SRB in the AMD solution, but not as low as the start pH of 2.0. Most of the AMD treatment plants by SRB run at a pH of about 7.0, some have also reported the experiment at the low start pH of 2.75 [25] and 3.67 [24]. Although under a pH of 2.0, the *Ferroplasma* and *Acidophilium* were the dominant species, some *Clostridiaceae* also showed resistance to the low pH and performed the capacity of sulfate-reduction. The SRB was not detected previously by the 16s rDNA-based high through-put sequencing in Monywa bioleaching heaps [19], so the SRB may have come from the acidic, anaerobic soil, that indicated the exogenous SRB might need the remediation of the heap leaching residue. AMD from the heaps was used for SRB cultivation; this will make the microbes more accustomed to the real industrial condition. Results proved that some SRB can grow in the very acidic environment, which provides the potential for the remediation of the acidic sulfide waste rocks.

### 3.2. Microbial Communities in Columns

The microbial community and microbial numbers during the incubation were tested in the pore solution at day 30, 90, 150 (Figure 2, Figure 3 and Figure 4, Table 4), and were also tested in the residue at the end of the tests (180 days) (Figure 5 and Figure 6), respectively. Organic matter and lime (to elevate pH) were added to the incubation columns for supporting the SRB and iron-reducing microbes to compete with the iron- and sulfur-oxidation microbes. It is shown that organic matter greatly promotes the growth of the SRB and the autotrophic bioleaching microbes were not detected anymore, and the addition of lime also inhibited the autotrophic acidophiles (Figure 2 and Figure 3).

For the microbial community in the pore solution, microbes in the control treatment were dominant by phylum Firmicutes and Nitrospirae, which are mainly the species of *Leptospirillum ferriphilum*, *Sulfobacillus*, some *Acidithiobacillus caldus*, and some *Alicyclobacillus ferrooxydans*. In addition, in the lime addition treatment, pH was elevated from 2.0 to about 3.9; the solution’s microbial community changed to the dominant of the phylum of Firmicutes and Bacteroidetes, with a high percentage species of *Hydrotalea flava*, and *Alicyclobacillus*, and also the growth of *Clostridiales*, with a small portion of *Sulfobacillus* and *Acidithiobacillus caldus*; however, *Leptospirillum* was not detected anymore. This suggested that *Leptospirillum* was not accustomed to the higher pH (at the end of the test, pH of 3.9), and the higher pH resulted in the growth of a small amount *Clostridiales*. Additionally, the treatment with organic matter showed a microbial community in the pore solution of *Spirochaetaceae bacterium SURF-1* (31.0%), *Ralstonia pickettii* (23.6%), *Geothrix* (12.5%), *Norank Family XVIII* (8.4%), and *Thiobacillus thioparus* (9.4%), while in the treatment with organic matter and lime together suggested the species of *Spirochaetaceae*
*bacterium SURF-1* (72.7%) and *Desulfurispora* (17.4%) in the incubation solutions. The addition of lime and organic matter together promotes the solution cell number (Figure 4). The species during the incubation at 30, 90, and 150 days suggested the dominance of sulfate-reducing bacteria of *Spirochaetaceae bacterium SURF-1* in the solution (Table 4). It is reasonable for the growth of *Spirochaetaceae bacterium SURF 1* since there is abundant sulfate in the residue, along with the organic matter input, the gradually increased pH, and the more anaerobic condition.

For the microbial community in the residue, microbes in the residue of the treatment Lime + OM promote the different SRB and iron-reducing groups (most in the phylum of Firmicutes), in the control column with Firmicutes, Nitrospirae, and Euryarchaeota, and in single lime or organic matter addition column with the predominance of Proteobacteria (Figure 5). Microbial community diversity in species in the residue after 180 days is much higher than that in the pore solution (Figure 6), suggesting that most of the microbes like to live in the residue rather than in the pore solution, while *Spirochaetaceae bacterium SURF-1* was more likely to exist in the solution, and in the residue, it was only in a proportion of about 10% (Figure 6) in the treatment of Lime + OM. In the residue of the control treatment, *Acidithiobacillus*, *Alicyobacillus*, *Sulfobacillus*, and *Ferroplasma* were the main genus. The microbial community in the treatment with lime was high in the genus of *Ralstonia*, and in the OM treatment, *Burkholderia*, *Clostridium*, *Thiobacillus*, and *Desulfosporosinus*. The most abundant and diversity of microbes were seen in the lime + OM treatment, with the high diversity of SRB groups (such as in *Clostridiaceae, Peptococcaceae*) and also in iron-reducing microbes such as *Geobacter, Geothrix, Therminocola*. The bacterial population also included other organisms such as fermenting bacteria and methanogens, some of which help to hydrolyze and ferment complex carbons to readily available substrates for the SRBs (Figure 6). It showed a higher diversity index of Shannon and Simpson in the Lime + OM treatment (Shannon index of 4.36 in column 4 with Lime + OM, while columns 1–3 of 2.18, 1.61, and 2.94, respectively; Simpson index of 0.92 in column 4 with Lime + OM, while columns 1–3 of 0.70, 0.30, and 0.75 respectively).

Anaerobic microorganisms were detected by culture techniques in mine dumps located in Thuringia, Germany, in Ontario, Canada or Montana, USA [3,17,26], and were quantifiable of the neutrophilic Fe (III)-reducing *Geobacteraceae*, as well as the *dsrA* gene of sulfate reducers at the tailing dumps at the pH > 3.5. And it also proved a highly varied and complex microbial ecology within a very heterogeneous geochemical environment [17]. It has also been reported that with populations of SRB, reduced sulfur species are produced, thus elevated populations of sulfur-oxidizing microbes were also found [16]. The iron and sulfur oxidation and reducing processes co-existed in the environment, but it was still suggested that the organic matter in the tailing dam or bioleaching heaps residue was too low for supporting enough SRB activity to reverse the oxidation process; thus, it is proven that the addition of organic matter is efficient for the growth promotion of SRB.

The ferric iron as an oxidant in the acid condition, together with the autotrophic acidophiles, provides favorable conditions for the dissolution of sulfide minerals [5]. It has been revealed that creating anoxic conditions would not only inhibit the activities of iron- and sulfur-oxidizing bacteria, but also would enhance activities of SRB and iron-reducing bacteria which could successfully be used in preventing and/or treating ARD [27]. For the first importance, the anaerobic condition inhibits the autotrophic acidophiles, created by input of the heap bioleaching residue using the water flood in the columns. In addition, acidophilic bacteria cannot utilize organic substances; some acidophilic bacteria can even be inhibited or killed by some organic substances [28]. By contrast, the growth of heterotrophic SRB depends entirely on complex organic substances and can be promoted by organic substances. Yeast extract, together with sodium lactate, provides balanced nutrition for the SRB rather than the single organic substance, which is also shown in [24] the efficiency of yeast powder, and also lots of natural organic matters are suitable for the promotion of SRB [29,30]. At the same time, the oxidation of organic matter coupled with the reduction of ferric and sulfate and generates alkalinity, while SRB favors higher pH. These all result in the growth of SRB and also the iron-reducing microbes during the incubation in sulfide waste rocks.

It can be summarized that, under the oxygen deficiency condition in the acidic sulfide waste rocks, species favor or can grow in an anaerobic environment; *Acidiphillum*, *Ferroplasma* and *Alicyclobacillus* will grow and compete with the aerobic acidophilic species, such as *Acidithiobacillus* and *Leptospirillum* [5]. As most anaerobic microbes rely on the organic substance, the organic substance input will further promote these anaerobic genera, while reducing Fe^3+^ to Fe^2+^ (such as *Ferroplasma*, *Acidophilium*), which will facilitate the pH increase. If the pH increases above 3, some acid-tolerant SRB groups, such as in family *Clostridiaceae*, then *Peptococcaceae,* will grow, and Fe^3+^ and SO_4_^2−^ reduction will happen at higher reaction rates and increase the pH continuously.

### 3.3. Physiochemical Properties in Different Treatments

The physicochemical characteristics in columns were significantly different between treatments (Figure 7). It was showing that the organic matter addition (OM, Lime + OM) finally resulted in the pH increase to nearly neutral, while the single lime addition did not finally increase the pH (Figure 7a), suggesting the importance of the organic matter, as also shown in the microbial community while the SRB was promoted greatly.

Additionally, during the incubation, the Fe^2+^ concentration in the organic matter addition treatments (OM, Lime + OM) showed an increased at about 40 and 70 days, respectively (Figure 7b), this time point was also correlated to the time when the pH started to increase, suggesting the competitive growth of the SRB and iron-reducing microbes at that time. Then, the Fe^2+^ concentration decreased, suggested the co-precipitate with the S^2−^. The control treatment also had a higher Fe^2+^ concentration in the beginning because the pH was lower, resulting in pyrite dissolution after the start of the incubation. The decrease of the redox potential in the organic matter addition treatments (OM, Lime + OM) was also at the same time the start of the iron- and sulfate-reduction (Figure 7c). With the reduction of Fe^3+^ to Fe^2+^ and SO_4_^2-^ to S^2−^, the Eh decreased significantly, while the control and lime addition treatments were quite stable, suggested the lesser reducing reaction. It was reported that below the Eh of about 450 mV (vs. Ag/AgCl), pyrite oxidation was negligible even it was fully covered by bioleaching microbes on the surface [31]. So, in all the four treatments, the pyrite oxidation was very slow as their Eh were all lower than 450 mV; thus the pH in the column was not decreased, showing the effectiveness of the anaerobic condition to inhibition the pyrite oxidation. Organic matter oxidation further consumed the dissolved oxygen, which further inhibited the autotrophic microbes. Dissolve organic carbon in the solution decreased during the inoculation (Figure 7d); maybe it was used by the heterotroph microbes for their growth.

The overall reducing processes resulted in an improvement in the pore water quality due to precipitation of metals as sulfides with the H_2_S generated in organic substrate and neutralization of the acidity due to the bicarbonate released during sulfate reduction. As pH increases, aqueous metal species tend to precipitate as hydroxide, oxyhydroxide, or hydroxy sulfate minerals. Also, dissolved metals may adsorb onto the surfaces of these secondary minerals and other surfaces present in the environment, and increase hydrolysis of metal ions at circum-neutral pH [32,33]. With the co-precipitation of the Cu together with the microbial-produced sulfide, the final Cu concentration in the solution of treatment with lime and organic matter together were below the detection limitation (<1 mg/L), while in the treatments of lime addition and organic matter were 56 and 26 mg/L, respectively. All of them were much lower than the control treatment of 400 mg/L (data not shown). This process is also particularly effective for removing heavy metals such as cadmium, copper, lead, mercury, zinc, and iron to low concentrations, as also previously reported [27].

Researchers have tried to use SRB to remediate the AMD in the last several years, but most were the ex situ treatment after the AMD had already come out from the sulfide waste rocks [13,14,15]. Here, the source control to inhibit the oxidation of sulfide minerals directly in the waste ore was carried out. The results showed the elevation of the pH after the promotion of the iron-reducing microbes and SRB groups. By inoculation of the local anaerobic SRB and the addition of organic matter, the pH can be increased to neutral. Ferric-reducing microbes also exist under very low pH and may be responsible for the initial process, and also the ferric reducing rates are higher than the sulfate reduction rates at the beginning of the remediation of the acidic waste rocks [34], so microbial alkalinity generation may first happen by iron reduction and then by both iron and sulfate reduction. The bioremediation of the acidic sulfide containing waste rocks is possibly first relying on the acidophilic iron-reduction to elevate the pH to about 3.0, and then the growth of the SRB and iron-reducing bacteria to precipitate the metals and sulfate, which result in the continual increase of pH. Even with small amounts of the iron-reducing and SRB microbes, their reducing reaction will consume the oxygen, reducing the Fe^3+^ and SO_4_^2−^, decreasing the Eh, and then benefiting the inhibition of the pyrite oxidation. Additionally, if these microbes become dominant, it will finally reverse the reactions causing acid mine drainage, producing alkalinity, attenuating the movement of metals by the precipitation of sulfide minerals, and, finally, bio-remediate the waste rocks (Figure 8). For the rehabilitation of acidic waste rocks, one layer of lime and organic matter will be useful to inhibit the microbial assisted sulfide mineral oxidation in the oxidation front of the waste rocks, and therefore provide a protection layer for upper plant growth. For the remediation of mining acidic lakes, SRB-combined organic matter input will help the neutralization and heavy metal precipitation in the lakes.

## 4. Conclusions

The consortiums of iron- and sulfate-reducing microorganisms in low pH acid mine drainage were cultured. Then, the consortiums were applied to the heap bioleaching residue from a copper mine to investigate the competitive growth of iron- and sulfate-reducing bacteria with the local autotrophic acidophiles under different conditions. It is proved that there is growth of the anaerobic heterotrophic microbes under anaerobic acidic conditions, mainly the iron-reducing microbes, such as *Acidiphillum*, *Ferroplasma,* and *Alicyclobacillus*, especially when the pH value is lower (pH < 3), and then this gradually turns to the growth of SRB and iron-reducing microbes together if the pH increases. If there is input of organic matter, the SRB and iron-reducing microbes will then be greatly promoted, such as in family *Clostridiaceae*, *Peptococcaceae*, *Spirochaetaceae,* and will result in the co-precipitation of the metals with biogenic sulfide, and also elevate pH, finally resulting in heavy metal precipitation in situ of the waste rocks.

This study gave implications to the remediation of the acid waste rocks. It is possible to trigger SRB growth in the waste pyritic rocks to inhibit the pyrite oxidation, and eventually reverse the reaction of sulfide mineral oxidation, then metal-sulfide co-precipitation in situ of the waste rocks. The SRB cultivation will not only be used ex-situ for the AMD attenuation, but also can be used in situ to inhibit pyrite oxidation, and will also help to form a layer of the hinder from the acid-generating if used for further rehabilitation of the waste rocks, or in-situ the acid mine drainage lake to help the neutralization and heavy metal precipitation.

## Figures and Tables

**Figure 1 ijerph-17-02715-f001:**
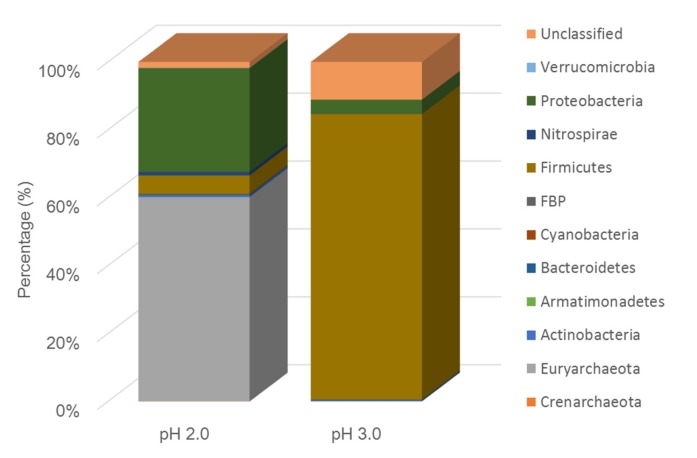
Phylum of the microbes after the anaerobic incubation at pH of 2.0 and 3.0, respectively.

**Figure 2 ijerph-17-02715-f002:**
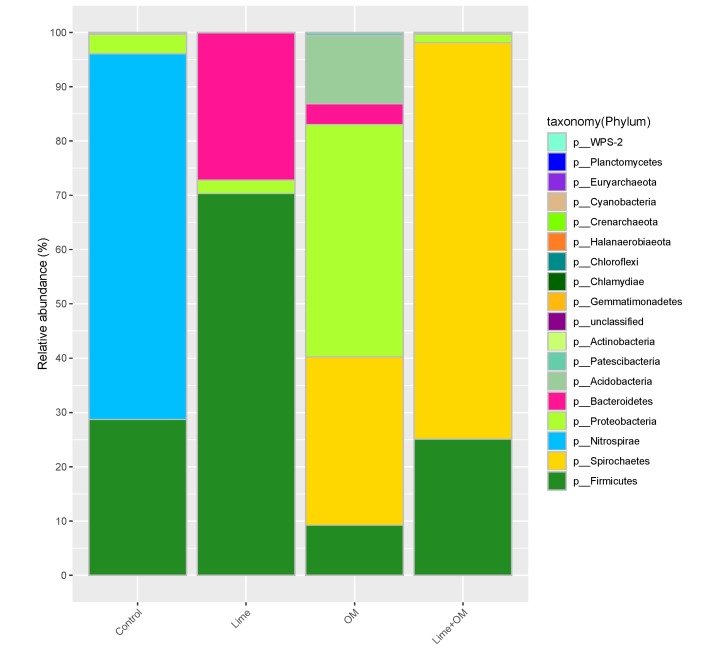
Phylum of the microbial community in the solution phase of the four columns (control, Lime, organic matter (OM), and Lime + OM) at day 150.

**Figure 3 ijerph-17-02715-f003:**
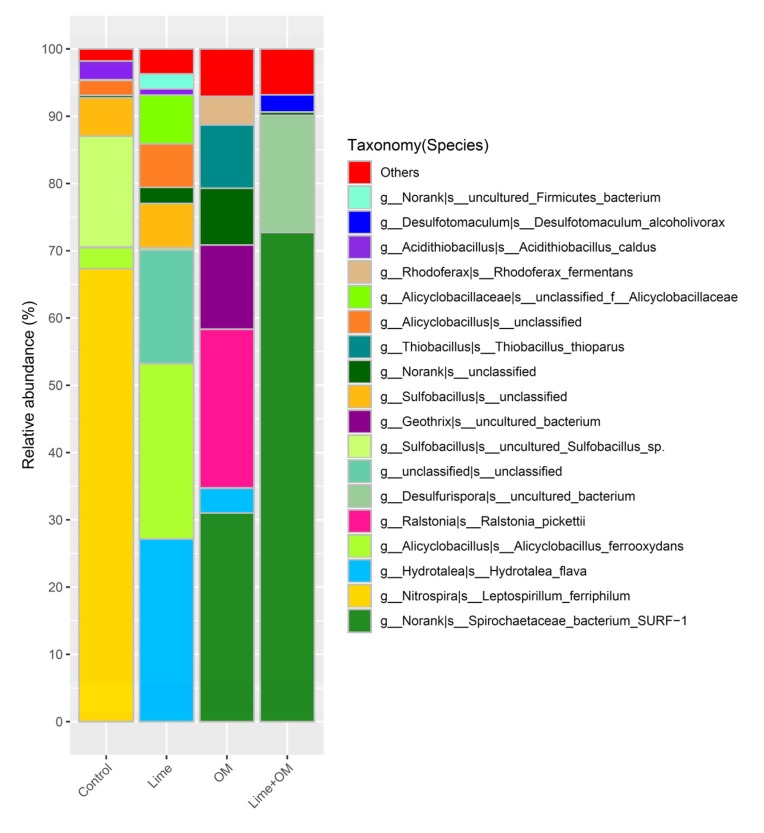
Species of the microbial community in the solution phase of the four columns (control, Lime, OM, and Lime + OM) at day 150.

**Figure 4 ijerph-17-02715-f004:**
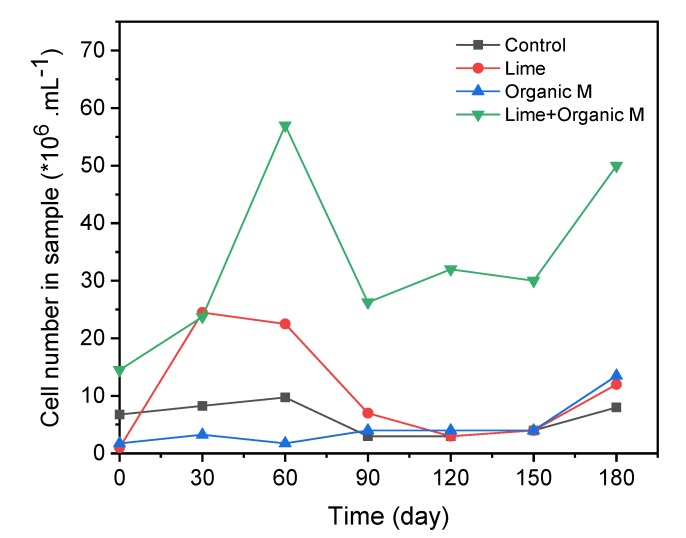
Cell number in the solution phase of the four columns (control, Lime, OM, and Lime + OM) during the incubation.

**Figure 5 ijerph-17-02715-f005:**
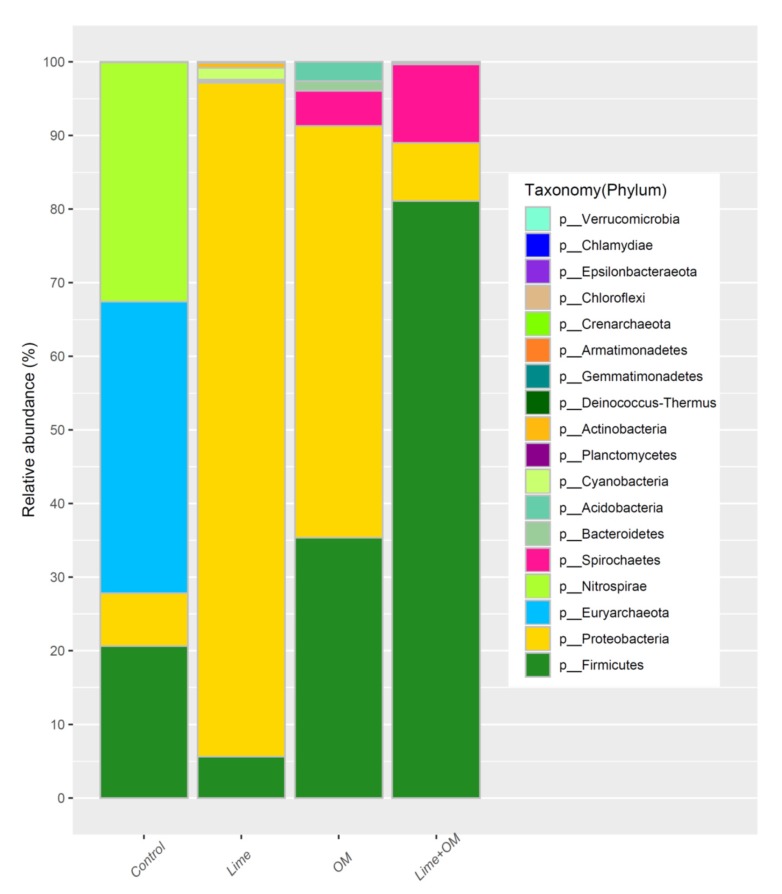
Phylum of the microbes in the residue of the four columns (control, Lime, OM, and Lime + OM) at the end of the inoculation of 180 days.

**Figure 6 ijerph-17-02715-f006:**
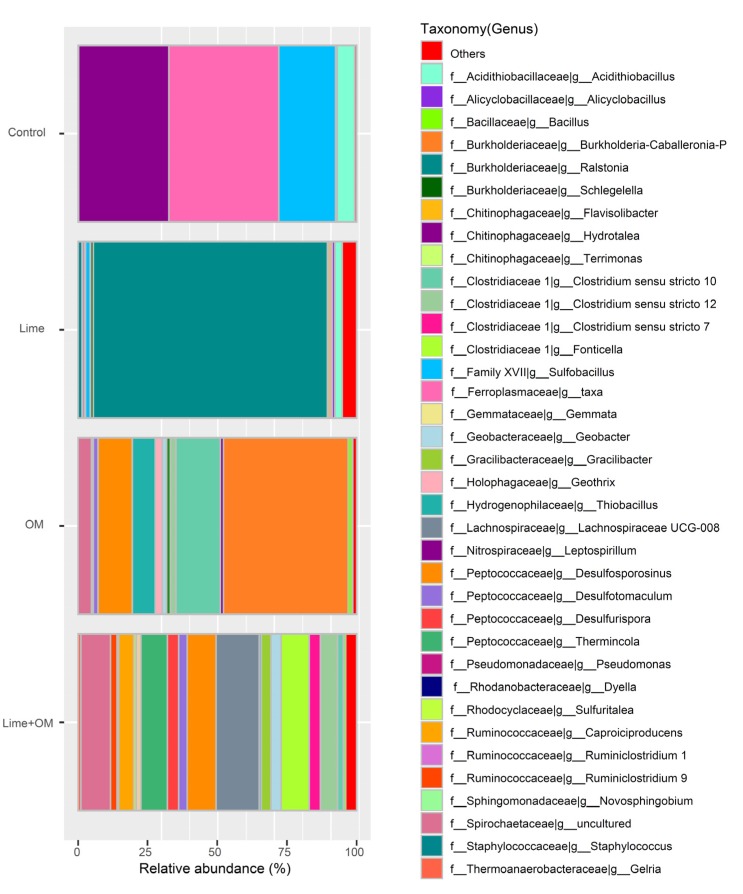
Genus of the microbial community in the residue of the four columns (control, Lime, OM, and Lime + OM) at the end of the incubation of 180 days.

**Figure 7 ijerph-17-02715-f007:**
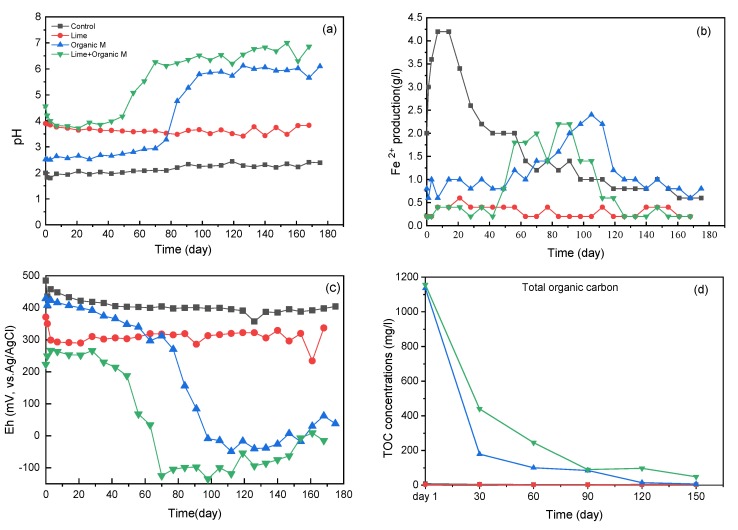
pH value (**a**), Fe^2+^ concentration(**b**), Eh (**c**), total organic carbon (**d**) in the solutions from the four columns during the experiment of the four columns (control, Lime, OM, and Lime + OM), respectively.

**Figure 8 ijerph-17-02715-f008:**
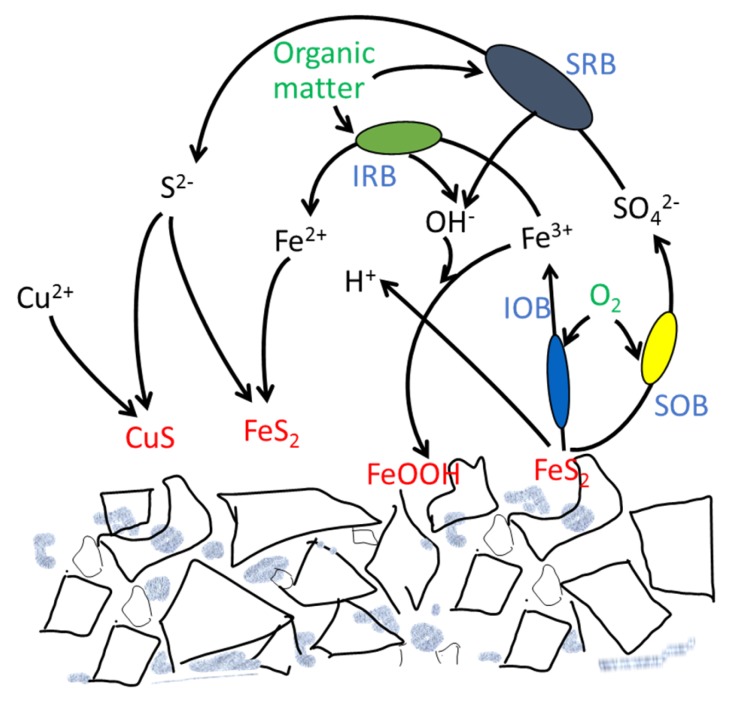
Scheme of the microbial-induced transformation of the minerals for bioremediation of pyritic waste rocks. Sulfide minerals, such as pyrite, are oxidized by IOB and SOB to Fe^3+^ and SO_4_^2−^ under aerobic conditions and form the acid mine drainage. After the creation of the anaerobic condition, and input of organic matter, the acidic IRB reduce Fe^3+^ to Fe^2+^ and elevate the pH, and then the less acid-tolerant SRB and IRB grow, thus the pH increases gradually, and, finally, the metal ions precipitate together with the microbial generated sulfide and with the elevated pH. SOB: sulfur oxidizing microbes, IOB: iron oxidizing microbes; IRB: iron reducing microbes; SRB: sulfur reducing microbes.

**Table 1 ijerph-17-02715-t001:** Element assay of the acid mine drainage during heap bioleaching.

**Fe mg/L**	**Cu mg/L**	**Ca mg/L**	**Mg mg/L**	**K mg/L**	**Na mg/L**
11,520	1644	630	1002	7.8	790
**Al mg/L**	**As mg/L**	**Cd mg/L**	**Zn mg/L**	**Pb mg/L**	**Mn mg/L**
1697	16.9	2.6	85.1	<0.1	271
**Hg mg/L**	**Cr mg/L**	**Co mg/L**	**Se mg/L**	**Si mg/L**	**Ag mg/L**
<0.1	0.8	8.6	<0.1	133	<0.1
**pH**	**Eh** **mV vs. Ag/AgCl**	**SO_4_^2−^ mg/L**	**Cl^−^ mg/L**	**Free Acid g/L**	**Total Ionic Strength %**
1.10	520	73,800	2018	9.21	8.08

**Table 2 ijerph-17-02715-t002:** Characteristics of the heap bioleaching residue.

Sample	Cu %	Fe %	Ca %	Mg %	Al %	Na %	K %	Total Sulfur %	Reduce Sulfur %	Net Acid Production Potential Kg/t
Cell 1	0.064	5.70	0.10	0.12	8.45	0.28	2.66	5.60	3.57	72.9
Cell 2	0.078	6.86	0.09	0.08	9.27	0.26	2.93	9.02	5.71	116.5
Cell 3	0.074	4.00	0.12	0.12	9.51	0.28	2.61	4.63	2.16	44.1
Average	0.070	5.52	0.10	0.11	9.08	0.27	2.73	6.42	3.81	77.8

**Table 3 ijerph-17-02715-t003:** Genus of the microbes and their percentage after the anaerobic incubation at pH of 2.0 and 3.0, respectively (percentage >2%).

pH 2.0	pH 3.0
*Ferroplasma*: 61.8%	*Clostridiaceae*: 67.9%
*Acidophilium*: 23.0%	*Desulfosporosinus*: 18.7%
*Clostridiaceae:* 5.5%	*Alicyclobacillaceae*: 8.6%
*Thermogymnomonas*: 4.7%	*Acidophilium*: 2.0%

**Table 4 ijerph-17-02715-t004:** Dominant species in the solution phase of organic addition columns (OM, Lime + OM) during the incubation in the column (abundance >5%).

Sample Time (Day)	Cl-3 (OM)	Cl-4 (Lime + OM)
30	*Alicyclobacillus ferrooxydans: 32.7%*	*Clostridium sensu stricto 10: 66.8%* *Desulfosporosinus acidiphilus: 27.2%*
	*Desulfosporosinus acidiphilus: 23.0%*	
	*Clostridium sensu stricto 10: 17.1%*	
	*Hydrotalea flava: 11.6%*	
	*Penicillium Janthinellum: 7.9%*	
90	*Hydrotalea flava: 21.3%*	*Spirochaetaceae bacterium SURF-1: 70.5%* *Desulfovibrio: 8.5%*
	*Geothrix: 13.5%*	
	*Ralstonia pickettii: 6.2%*	
	*Desulfotomaculum: 5.5%*	
	*Unclassified Anaerovorax: 9.4%*	
	*Pseudolabrys: 8.3%*	
	*Desulfotomaculum: 6.6%*	
150	*Spirochaetaceae bacterium SURF-1: 31.0%*	*Spirochaetaceae bacterium SURF-1: 72.7%* *Desulfurispora: 17.4%*
	*Ralstonia pickettii: 23.6%*	
	*Geothrix: 12.5%*	
	*Norank Family XVIII: 8.4%*	
	*Thiobacillus_thioparus: 9.4%*	

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
