# Peer review of "Competitive Growth of Sulfate-Reducing Bacteria with Bioleaching Acidophiles for Bioremediation of Heap Bioleaching Residue"

_ijerph, 2020, doi:10.3390/ijerph17082715_

Round 1

Reviewer 1 Report

The paper looks at the competitive growth of sulfate reducing bacteria with
bioleaching acidophiles for bioremediation of heap bioleaching residue.

The paper requires a through proof read for English.

The authors should give context to how big a problem mine waste of this type is environmentally, economically, etc and how what this paper layout may help to deal with it.

Where do the figures presented in Table and Table 2 come from?

Fix the font on bacterial names throughout the text.

In Line 102 the authors state that "Fe2+ concentration in the bottles was timely measured". What do the authors mean here by "timely".

In Line 103 replace "was measured" with "was analyzed"

Line 105 makes no sense, please rewrite.

In Line 108 the authors state "Subsequently, its size distribution and Cu content for each size were assayed" How were these analyzed?

Line 191 replace "The former researches also" with "Previous studies"

Line 233 Should "Rostonia" be "Ralstonia"?

Do the authors have any speculation on why the microbial ecology changed throughout the 150 days (especially on why Ralstonia pickettiii and Spirochaetaceae bacterium SURF-1 became so dominant?)?

The paper is interesting and with some corrections I think it will be publishable. 

Author Response

The paper requires a through proof read for English.

Response: Thanks for the review of the manuscript and the useful suggestions. The English is revised by a native English speaker. Revision can be seen in the manuscript with trace of changes.

The authors should give context to how big a problem mine waste of this type is environmentally, economically, etc and how what this paper layout may help to deal with it.

Response: We add the sentences in paragraph 4 to descript this: “Mining activities generated waste include waste rock, tailings, and heap leaching residue. All around the world, acid-generating wastes were produced about billions of tons every year, and the generation of acid mine drainage have been the most important resources of the heavy metal pollution all around the world. Among the acid-generating wastes, the heap leaching residue was generate every year by hundreds of millions tons, and is the most difficult to remedy and to revegetate, because during the heap bioleaching process, the whole heap is completely acidified, usually under the pH<1.5, and also the growth of iron- and sulfur- oxidation microbes is promoted during the bioleaching process.”

Where do the figures presented in Table and Table 2 come from?

Response: Thanks! We add the test method of the data: “Leach solution from the bottom of the cell is collected, and the cation ions were assayed by atomic absorption spectrometry (AAS) and inductively coupled plasma mass spectrometry (ICP-MS), and anion ions were assayed by ion chromatography (IC), free acid was assayed by titration, and ionic strength was assayed by precipitate all elements and got the total weight (Table 1)”, and “The element assay by ICP-MS and net acid generating potential was assayed by H2O2 oxidation and NaOH consumption.”.

Fix the font on bacterial names throughout the text.

Response: Thanks! We revised throughout the manuscript.

In Line 102 the authors state that "Fe2+ concentration in the bottles was timely measured". What do the authors mean here by "timely".

Response: We measure the Fe2+ weekly, so we change “timely” to “weekly”.

In Line 103 replace "was measured" with "was analyzed"

Response: Thanks! We replaced it.

Line 105 makes no sense, please rewrite.

Response: Thanks! We rewrote the sentence: “And the two microbial consortiums after cultivation at pH 2.0 and 3.0 were used as the inoculation SRB in the column tests.”.

In Line 108 the authors state "Subsequently, its size distribution and Cu content for each size were assayed" How were these analyzed?

Response: Thanks! We add the information: “Subsequently, its size distribution were assayed by standard sieves (10, 5, 2, 1, 1/2 inches, 100 mesh and 200 mesh) and Cu content for each size were assayed.”

Line 191 replace "The former researches also" with "Previous studies"

Response: Thanks! We changed it.

Line 233 Should "Rostonia" be "Ralstonia"?

Response: Thanks! We changed it.

Do the authors have any speculation on why the microbial ecology changed throughout the 150 days (especially on why Ralstonia pickettiii and Spirochaetaceae bacterium SURF-1 became so dominant?)?

The paper is interesting and with some corrections I think it will be publishable. 

Response: Thanks! We added some discussion on this: “The species during the incubation at 30 days, 90 days and 150 days suggested the dominance of sulfate reducing bacteria of Spirochaetaceae bacterium SURF-1 in the solution (Table 4). It is reasonable for the growth of Spirochaetaceae bacterium SURF 1, since the abundant sulfate in the residue, the organic matter input, the gradually increased pH and more anaerobic condition. Spirochaetaceae bacterium SURF-1 is more like to exist in the solution, and in the residue only in a portion of only about 10% (Figure 6) in the treatment with lime and organic matter together.”.

Reviewer 2 Report

The paper present the results of study which investigated the heap bioleaching residue used for the bioremediation process by sulfate-reducing bacteria and the autotrophic bioleaching microbes.

Before acceptance, I strongly suggest to check all technical aspects of the paper. There is various fonts in the text, the reference number and order of their appearance should be checked (see atached file).

I suggest authors to prepare and incorporate in text block scheme or graphical presentation of the examined bioremediation process.

Author Response

Thanks a lot for your useful suggestion !

(1) Before acceptance, I strongly suggest to check all technical aspects of the paper. There is various fonts in the text, the reference number and order of their appearance should be checked (see atached file).

Response: Thanks! We have checked throughout, and revised the font and references format, and revise according to the reviewer’s suggestion in the attached file.

(2) I suggest authors to prepare and incorporate in text block scheme or graphical presentation of the examined bioremediation process.

Response: Thanks! It is a good suggestion. We add the Figure 8 to showing the processes related to the bioremediation.

Reviewer 3 Report

My corcern is directed to reverisbility of the prosess that is ledind to neutralisation of the environment in your study. Are there any informations about domination abilities of bacterial communities that are key players in two different environmental conditions in terms of pH changes. I am afraied that it can be permanet irreversible change. Can you please find an appropriate expanation for my ceoncern related to change in biodiversity of bacterial communities in the environment you studied.

Author Response

Thanks for the comments. It is a good question! Bioremediation is an environment safe process. All the SRB microbes are natural existing in nature, so it will not a safety problem for the microbial community changes. Mining activity actually the change for the environment microbial community, which promote the growth of the acidophile, while the bioremediation is a process to made it more close to the environment before the mining activity.

Round 2

Reviewer 3 Report

I have only one more remark related to diversity of microorganisms. Results in Table 4 are not easy to uderstand and some explenation regarding low species diversity should be included. How it is possible that the genus number identified in Table 6 is much higher for different bioremediation methods
than the number of few spiecies included in Table 4? Is is for example related to the number of reference sequences you used for identification for different levels of microorganisms classification.

Author Response

Dear reviewer:

Thanks for your important suggestion. I think maybe we do not explain the data clearly enough. Actually the microbial community in table 4 is from the pore solution from the column (30day, 90day, and 150day), while in Figure 6 is from the final residue (180 day), so in the solution the diversity is much lower than that in the residue.

So we add the sentence in paper line 215 to make it more clear: “Microbial community during the incubation were tested in the pore solution at 30, 90, 150 day (Figure 2, Figure 3, Table 4), and were also tested in the residue at the end of the tests (180 day) (Figure 5, Figure 6), respectively.” And also some discussion about the diversity difference in line 243-247 “Microbial community diversity in species in the residue after 180 days is much higher than that in the pore solution (Figure 6), suggested that most of the microbes liked to live in the residue rather than in the pore solution, while Spirochaetaceae bacterium SURF-1 was more likely to exist in the solution, and in the residue was only in a portion of only about 10% (Figure 6) in the treatment of lime+OM.” And also compare the diversity between treatments in line 255-258: “It shown a higher diversity index of Shannon and Simpson in the lime+OM treatment (Shannon index of 4.36 in column 4 with lime+OM, while columns 1-3 of 2.18, 1.61, 2.94 respectively; Simpson index of 0.92 in column 4 with lime+OM, while columns 1-3 of 0.70, 0.30, 0.75 respectively).”

And we also made some small revisions, details can be seen from the manuscript with trace of changes.

Best regards!

Yours

Yan Jia